# Individual behavioral trajectories shape whole-brain connectivity in mice

Jadna Bogado Lopes[1,2], Anna N Senko[1,2], Klaas Bahnsen[3], Daniel Geisler[3], Eugene Kim[4], Michel Bernanos[4], Diana Cash[4], Stefan Ehrlich[3,5], Anthony C Vernon[6,7]*, Gerd Kempermann[1,2]*

[1]German Center for Neurodegenerative Diseases (DZNE) Dresden, Dresden, Germany; [2]Center for Regenerative Therapies Dresden (CRTD), TU Dresden, Dresden, Germany; [3]Division of Psychological and Social Medicine and Developmental Neurosciences, Faculty of Medicine, Dresden, Germany; [4]Department of Neuroimaging, Institute of Psychiatry, Psychology and Neuroscience King's College, London, United Kingdom; [5]Department of Child and Adolescent Psychiatry, Faculty of Medicine, Eating Disorder Treatment and Research Center, Dresden, Germany; [6]Department of Basic and Clinical Neuroscience, Institute of Psychiatry, Psychology and Neuroscience, King's College, London, United Kingdom; [7]MRC Centre for Neurodevelopmental Disorders, King's College, London, United Kingdom

**Abstract** It is widely assumed that our actions shape our brains and that the resulting connections determine who we are. To test this idea in a reductionist setting, in which genes and environment are controlled, we investigated differences in neuroanatomy and structural covariance by ex vivo structural magnetic resonance imaging in mice whose behavioral activity was continuously tracked for 3 months in a large, enriched environment. We confirmed that environmental enrichment increases mouse hippocampal volumes. Stratifying the enriched group according to individual longitudinal behavioral trajectories, however, revealed striking differences in mouse brain structural covariance in continuously highly active mice compared to those whose trajectories showed signs of habituating activity. Network-based statistics identified distinct subnetworks of murine structural covariance underlying these differences in behavioral activity. Together, these results reveal that differentiated behavioral trajectories of mice in an enriched environment are associated with differences in brain connectivity.

*For correspondence:
anthony.vernon@kcl.ac.uk (ACV);
gerd.kempermann@dzne.de
(GK)

Competing interest: The authors declare that no competing interests exist.

## Editor's evaluation

This important work is of broad interest to readers studying brain plasticity, individuality, and shared/non-shared environments. The identification of distinct patters of brain networks, in the absence of main effects, between two broad classes defining how mice explore their environments, is especially interesting. The evidence supporting their conclusions is convincing.

## Introduction

The 'non-shared environment' (**Plomin and Daniels, 2011**), the elusive component of the non-genetic factor in phenotypic variation (**Morgante et al., 2015**), contributes to interindividual differences and neurobiological individuality. In humans, in vivo neuroimaging studies have demonstrated a complex relationship between behavioral performance and brain structure associated with learning (**Draganski et al., 2014Taubert et al., 2010**), personality traits (**Nostro et al., 2017**), and political orientation (**Kanai et al., 2011**), supporting the idea of a causal, yet individual relationship between brain function,

**eLife digest** An individual's experiences and behavior shape their brain, thereby building and refining a network of connections between neurons. This unique network may affect an individual's brain resilience in the face of aging, injury or disease. Understanding how individual experiences shape brain connections could help scientists develop personalized treatments. It may also have important implications for preventing brain disease.

Studying mice can provide a window into some of these brain processes. By using inbred mice, scientists can rule out the role of genetics in brain differences. Scientists can also control the animals' environments and track the activity of individuals to study their behavior.

Bogado Lopes et al. show that more active mice living in enriched environments have signs of more complex networks of brain connections. In the experiments, the researchers placed genetically identical mice in either standard laboratory mouse housing or in enriched environments. Mice in the enriched housing had access to multi-level enclosures connected with tubes and supplied with a rotating array of toys. A tiny tracking device was inserted under the skin of the mice to follow their movements. Finally, all mice underwent structural magnetic resonance imaging to assess their brain anatomy and connections. This revealed that the most active and adventurous mice in the enriched enclosures had the most robust signs of increased brain connectivity. However, mice with declining activity levels in the enriched enclosures had fewer brain connections. Brain connection patterns in these creatures of habit were nearly identical to the ones in mice housed in small unenriched enclosures.

The results show that how individual mice respond to their environments affects their brain structure, and behavior. More active behavior patterns lead to more robust networks of brain connections. Larger studies in mice could provide more about lifestyle-dependent brain resilience. It may also help scientists to develop individualized approaches to optimizing brain health.

structure, and behavior. The structural connectome, which can be estimated based on the correlation between regional brain volumes (across subjects) or white matter links between specific brain regions (within subjects) partially recapitulates known functional networks and represents an individual 'fingerprint' in humans and non-human animals (*Alexander-Bloch et al., 2013*; *Melozzi et al., 2019*; *Pagani et al., 2016*; *Yee et al., 2018*). Open questions about the exact nature of the connectome and how it is represented in covariance patterns as assessed by imaging studies abound, especially with respect to the crucial issue of causes and consequences.

We have reported that environmental enrichment (ENR) increases variability in activity-related behaviors (roaming entropy, RE), or total object exploration in a novel object recognition task (*Körholz et al., 2018*). The ENR mice also showed greater variability in measures linked to brain plasticity, such as adult hippocampal neurogenesis (*Körholz et al., 2018*) accompanied by distinguishing epigenetic patterns in the hippocampal dentate gyrus (*Zocher et al., 2021*; *Zocher et al., 2020*). From these results we have developed the Individuality paradigm that exposes the influence of the non-shared environment on phenotypic variation (*Kempermann, 2019*; *Kempermann et al., 2022*). Its key feature is that both genes and environment are kept constant, while behavior is continuously measured to assess the emergence of differential behavioral trajectories.

Structural magnetic resonance imaging (sMRI) studies of mice exposed to different enrichment paradigms have confirmed historical *post-mortem* findings, such as increased hippocampal volumes (*Zhang et al., 2018*), and extended these to mouse brain regions involved in sensorimotor processing (*Scholz et al., 2015*). We here go a decisive step further and examine the relationships between individual behavioral trajectories and regional brain volumes as well as whole-brain structural *networks* ('connectomics'). These data may provide neurobiological foundations for concepts such as cognitive reserve or brain maintenance, which attempt to capture individual differences in healthy cognitive aging and resilience to neurodegenerative disease (*Kempermann, 2019*; *Scholz et al., 2015*).

# Results

We longitudinally tracked the behavior of enriched female mice (*n* = 38) in our Individuality cage system, equipped with radio-frequency identification (RFID) technology (*Evans, 2013*; *Figure 1A*). During 12 weeks of exposure to the ENR, mean values of RE as a measure of exploration and territorial coverage (*Freund et al., 2013Zocher et al., 2020*) were calculated per night and aggregated across four time blocks of 21 nights each. Increasing interindividual component of variance (*Figure 1B*) confirmed the emergence of individual behavioral differences. Two distinguishable patterns were observed (*Figure 1C*): mice with consistently high levels of RE and those with habituation and decreased RE values over time. Mice were stratified according to this criterion and grouped into 'flat' roamers (*n* = 15) or 'down' roamers (*n* = 15) depending on the slope of the linear regression line through a set of four RE time blocks. The slopes for 'flat' roamers ranged from −0.003 to 0.004, while slopes lower than −0.006 (to −0.049) were considered to represent 'down' roamers. Mice that showed in-between slope values (*n* = 2) and those with positive slopes (*n* = 6) were excluded from the clustering to achieve a sharper distinction between the extremes and equally sized groups of 15 mice were formed. The actual slopes are shown in Figure 3B. This is stratification serves the purpose of allowing the visualization of patterns at the top of a rank order based on changes in RE vs. at the bottom.

The number of doublecortin (DCX)-positive cells in the dentate gyrus, a proxy measure of adult neurogenesis, increased in ENR compared to standard conditions, but did not differ between the 'flat' vs. 'down' ENR subgroups (*Figure 1D*). As previously shown (*Evans, 2013*; *Gong et al., 2012*), individual variability in behavior positively correlated to adult hippocampal neurogenesis ($R^2$ = 0.20, p = 0.0046; *Figure 1E*). The positive correlation between end-point behavior (RE at the time block 4) and immature neurons was statistically significant in the 'down' roamers subgroup ($R^2$ = 0.40, p = 0.012), but missed conventional statistical significance in the 'flat' roamers ($R^2$ = 0.20, p = 0.095; *Figure 1F, G*). However, the correlations in the flat and down subgroups were not statistically different (one-sided Fisher's *z* test, *z* = −0.63, p = 0.27), which might point to an insufficient power for this comparison.

We next explored neuroanatomical changes between ENR- and standard-housed (STD) mice using ex vivo sMRI. Total brain volumes did not differ significantly (STD: 450.3 ± 11.7 mm³ vs. ENR: 456.5 ± 16.0 mm³; *t* = 1.14, df = 46; p > 0.05; q > 0.05; *Figure 2—figure supplement 1*). Using atlas-based segmentation (ABS), statistically significant (5% false discovery rate [FDR]) group differences in absolute volume (mm³) were found for 12% (22/182) of mouse brain atlas regions of interest (ROIs) with effect sizes (SMD) ranging from +2.7 in CA1 oriens (CA1Or) to +1.15 in the molecular layer of the dentate gyrus (*Figure 2A*; *Supplementary file 1*). Most of these ROIs (68%; 15/22) were located in the mouse hippocampus, but absolute volume increases were also observed in the infralimbic cortex (cingulate cortex area 25), olfactory nuclei, the anterior commissure, and the orbital cortex (*Figure 2A*; *Supplementary file 1*). The majority of these ROIs were conserved as significant group differences when considering relative volumes (% of whole brain), suggesting normal scaling (*Figure 2—figure supplement 2a*; *Supplementary file 1*). Complementing the atlas-based analysis, voxel-wise assessment of volume changes using tensor-based morphometry (TBM) revealed statistically significant (family-wise error [FWE] rate p < 0.05) clusters of voxels with apparent increases in both absolute volumes (*Figure 2—figure supplement 2b*) and relative volumes (*Figure 2—figure supplement 2c*) localized within the mouse hippocampus when comparing ENR to STD mice. Collectively, these data confirm the expected effects of ENR on mouse brain structure, as evidenced by prior mouse neuroimaging (*Scholz et al., 2015*; *Zhang et al., 2018*) and historical *post-mortem* studies in rats (*Diamond et al., 1966*).

We next explored to what extent our key behavior metric of RE relates to local volume changes (as measured by sMRI) in the ENR group alone. Using the average RE slope for each individual mouse in the ENR group across all four time blocks as a continuous variable, we performed a voxel-wise correlation of this measure against log-Jacobian measures of local volume change. This analysis revealed that RE slope values correlated positively with clusters of local volume changes in the mediodorsal and parafascicular thalamic nuclei, the periaqueductal grey, medial geniculate nucleus, deep mesencephalic nuclei, gray and white matter layers of the superior colliculus, and the central nuclei of the inferior colliculus (FWE p < 0.05, *Figure 2B*). At an exploratory threshold of p < 0.01 uncorrected for multiple comparisons additional positive correlations were seen between RE slope and the volumes of the sensory and motor cortices, striatum, ventral thalamus, brainstem nuclei, and cerebellar white

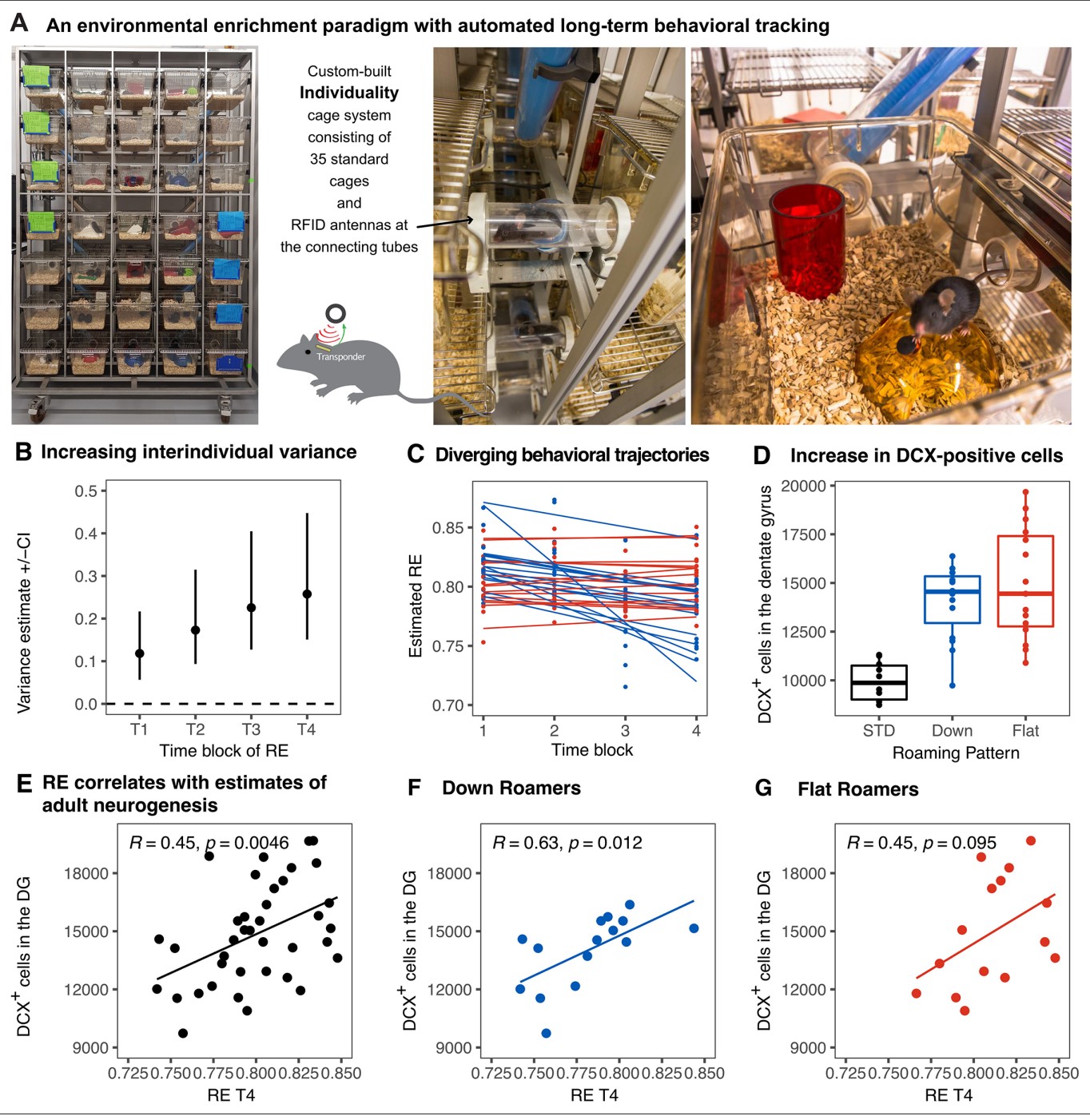

**Figure 1.** Emergence of inter-individual differences after environmental enrichment. Pictures (**A**) of the "individuality" cage—a custom build RFID cage with 70 interconnected small cages and 115 antennae distributed in the connecting tunnels. Interindividual variance (**B**) of behavior increased over time. Subgrouping of mice (n = 15) based on the slopes of RE trajectories (**C**), blue: 'down', and red: 'flat'. Animals living under enriched environment conditions (ENR) showed increased adult hippocampal neurogenesis (as assessed by proxy marker DCX) independent of behavioral trajectories (One-way ANOVA: $F_{(2,37)} = 17.3$, $p < 0.001$; Tukey post-hoc: standard vs flat and standard vs down $p < 0.01$, flat vs down $p = 0.46$) (**D**). Correlations between the number of doublecortin positive cells in the dentate gyrus to RE values at time-block four in the enriched group ($R^2 = 0.20$, $p = 0.005$) (**E**), down subgroup ($R^2 = 0.39$, $p = 0.012$) (**F**), and flat subgroup ($R^2 = 0.20$, $p = 0.095$) (**G**). Box and whisker plots: center line - median; upper and lower hinges - first and third quartiles; whiskers - highest and lowest values within 1.5 times the interquartile range outside hinges; dots - individual data points.

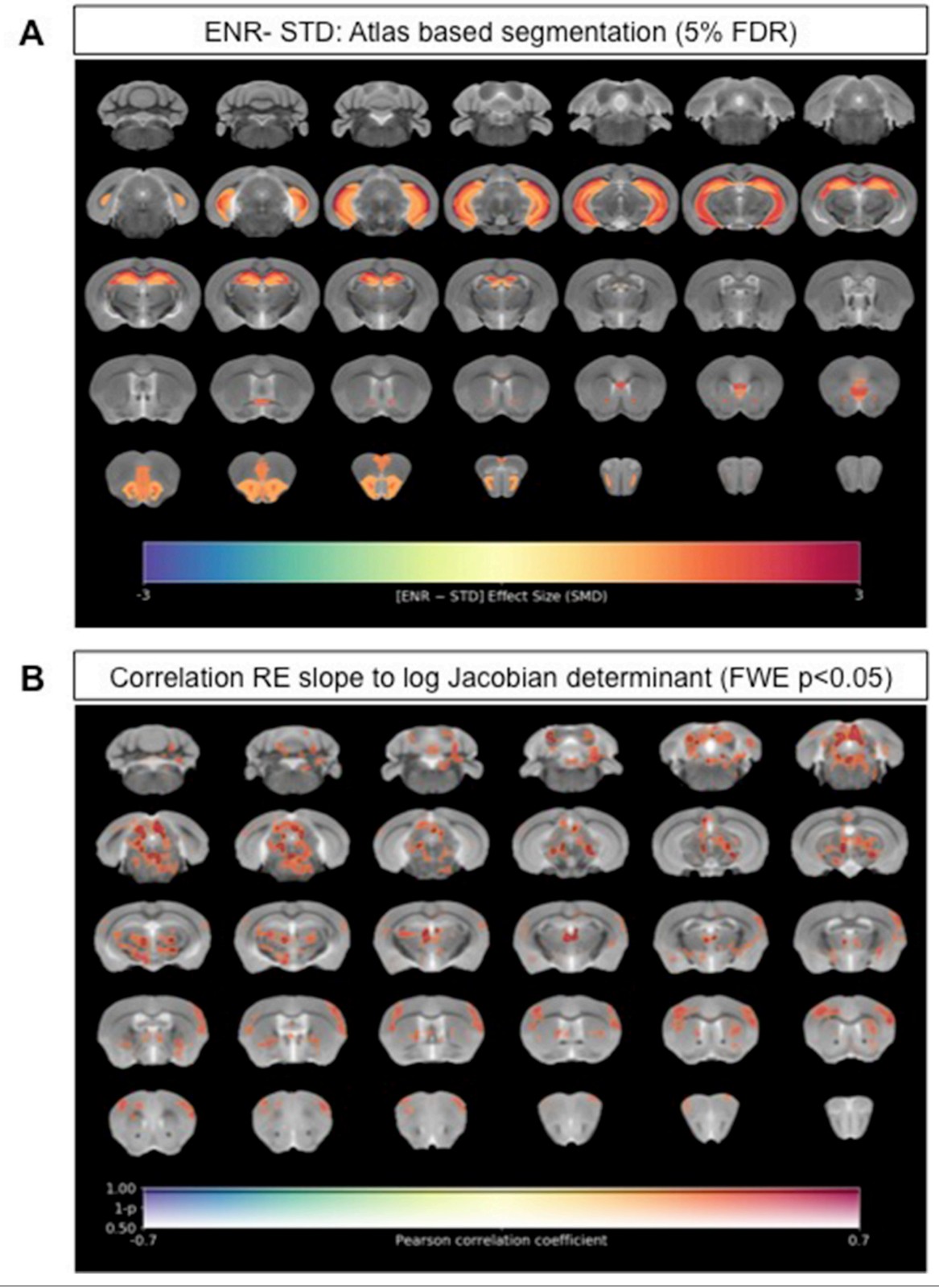

**Figure 2.** Neuroanatomical differences associated with exposure to an enriched environment and with variation in roaming entropy as measured by ex vivo structural MRI. (**A**) Map of regional differences in absolute volume (mm3) extracted using the DSURQE mouse brain atlas comparing ENR mice (N=38) to those in standard (STD) housing (N=10). Only atlas regions of interest (ROI) that survive multiple comparisons correction are shown (False Discovery Rate, 5%). Data shown for each statistically significant ROI is the standardized mean difference (SMD) comparing ENR – STD housed mice.

*Figure 2 continued on next page*

*Figure 2 continued*

Warm colors indicate regional volume increases, whilst cold colors indicate regional volume decreases in ENR-mice as compared to those in STD housing. (**B**) Positive correlation (Pearson's *r*) between roaming entropy slope across all four time blocks and local brain absolute volume changes (log-scaled Jacobian determinant) for all mice in the ENR group (N=38). Voxel clusters with solid contours are significantly correlated to behavior at threshold of p<0.05 (FWE corrected) and voxels without contours correlate at p<0.001 uncorrected for multiple comparisons. Warm colors indicate positive correlation, such that local volume increases are seen in mice with higher RE scores.

The online version of this article includes the following figure supplement(s) for figure 2:

**Figure supplement 1.** Total brain volumes (as measured by ex vivo structural magnetic resonance imaging [MRI]) do not differ significantly (2-tailed students t-test) when comparing either (**a**) STD (N=10)- vs. ENR-housed mice (N=38) or (**b**) Down (N=15) vs. Flat (N=15) subgroups of ENR-housed mice.

**Figure supplement 2.** Neuroantomical differences comapring enriched and standard housed mice as measured by ex vivo structural MR.

**Figure supplement 3.** Voxel-wise tensor-based morphometry (TBM) to map localized neuroanatomical differences between Flat and Down subgroups of ENR-housed mice.

**Figure supplement 4.** Relationship between RE and local volume changes in Down and Flat roaming mice.

matter. These data suggest that in mice with greater RE behavior in the ENR group the absolute volumes of these regions were larger than in the mice that displayed less RE behavior.

We next explored whether the subgroups within the ENR group that reflected continued high vs. declining ('flat' vs. 'down' roamers) differed anatomically, as measured by sMRI. Total brain volumes did not differ between the two ENR subgroups ('down': 451.7 ± 11.8 mm$^3$ vs. 'flat': 459.2 ± 5.2 mm$^3$; $t$ = 1.24, df = 28; p > 0.05; q > 0.05; *Figure 2—figure supplement 1*). Voxel-wise TBM to map localized neuroanatomical differences between Flat and Down subgroups of ENR-housed mice (*Figure 2—figure supplement 3*) did not show statistically significant localized volume differences between the groups after a stringent correction for multiple comparisons (FWE p < 0.05). We additionally ran a voxel-wise regression between the RE slope across all four time blocks against local volume changes for each individual mouse in the 'flat' and 'down' subgroups. In contrast to the correlation across all mice in the ENR group, there were no statistically significant correlations after correction for multiple comparisons (FWE p < 0.05). At an exploratory threshold of p < 0.01 uncorrected for multiple comparisons, however, RE slope values correlated positively with the absolute volumes of the cingulate, motor, somatosensory, insular, visual, and auditory cortices, as well as ventral thalamic, mid and hindbrain nuclei (*Figure 2—figure supplement 4*) largely replicating the pattern in *Figure 2B*. Collectively, these data suggest that individual differences in RE behaviors are likely supported by widely distributed mouse brain circuitry.

To study whether the emergence of interindividual variation in RE was associated with increased variability in the volumes of mouse brain regions, we calculated the coefficient of variation (CV) for each individual brain region. To determine if there are different degrees of overall variability between ENR subgroups, we averaged the CVs across all 182 ROIs in the mouse brain atlas to yield an average variability measure for each group (*Figure 3—figure supplement 1*). Comparing the sum of ranks between 'down' and 'flat' mice, we found a highly statistically significant difference (Mann–Whitney *U* = 11,076; p < 0.0001; *Figure 3—figure supplement 1*). These data are indicative of an increase in regional brain volume variance as a function of behavioral response to the enriched environment.

Structural covariance is defined as the correlated variation in volumes between pairs of brain regions, which, as human neuroimaging studies suggest, reflects both structural (*Gong et al., 2012*) and functional brain connectivity (*Segall et al., 2012*). Structural covariance networks are conserved in the mouse brain, providing an opportunity to explore the impact of ENR compared to standard housing on this cross-species measure of brain connectivity (*Pagani et al., 2015*; *Yee et al., 2018*). Correlation matrices of mouse brain structural covariance (*Figure 3A*) revealed a broad effect of housing, which was found to be statistically significant (Chi-square tests for equality of two correlation matrices; Chi-square = 17,997.7, df = 16471, p < 0.0001, where *prob* is the probability of observing the Chi-square under the null hypothesis) (*Steiger, 1980*). Furthermore, comparison of structural covariance matrices within the ENR group revealed distinctive effects between 'down' and 'flat' roamers (Chi-square = 19,997.29, df = 16,471, p < 0.0001; *Figure 3B*). The structural covariance matrix of 'flat' roamers (*Figure 3C*) differed from that of STD animals (Chi-square = 17,938.04, df = 16,471, prob <0.0001), whereas 'down' roamers (*Figure 3D*) were highly similar to STD group (Chi-square = 14,327.89, df = 16,471, prob <1; *Figure 3A*). In general, brain regions in the 'flat' roamers were

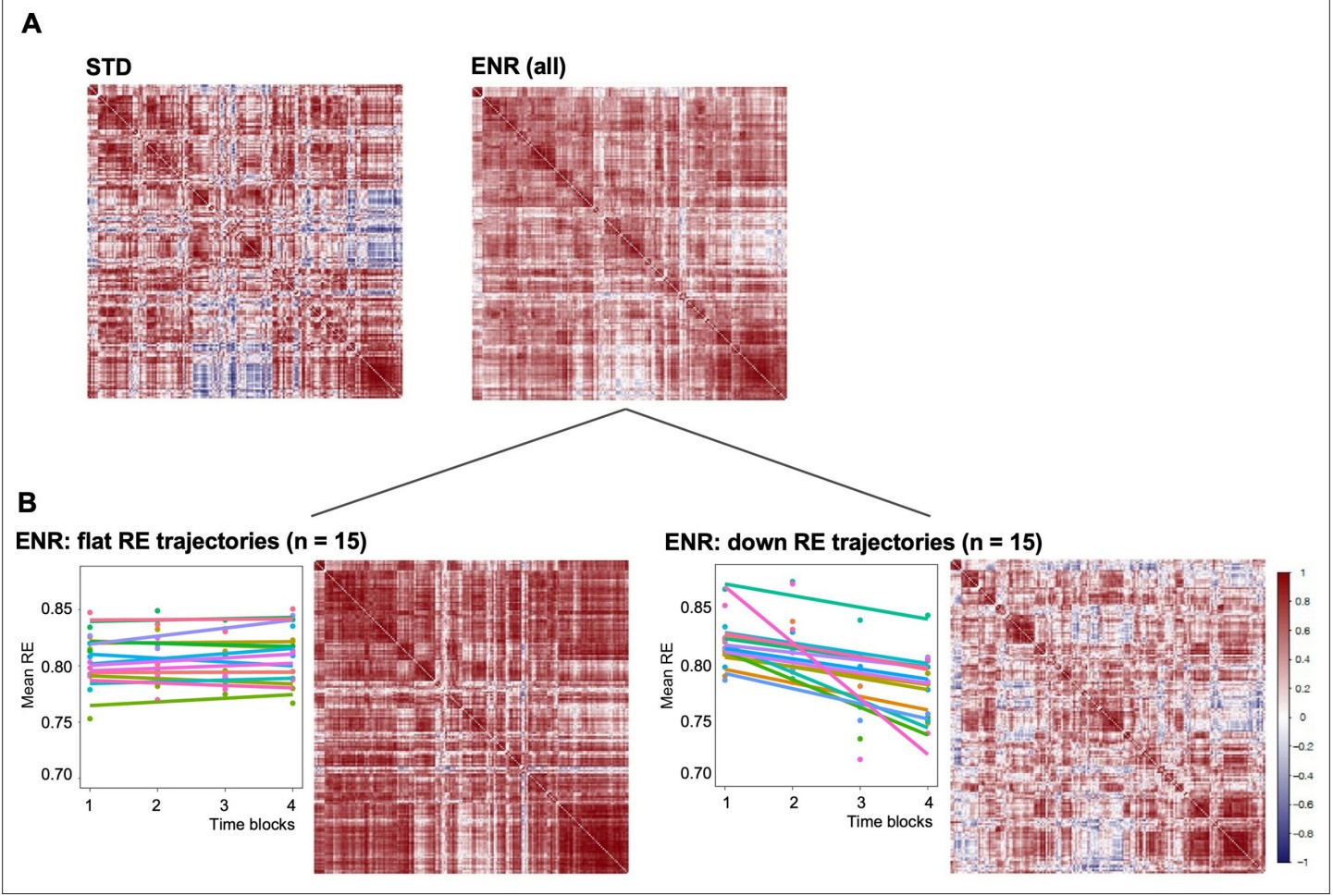

**Figure 3.** Environmental enrichment distinctively alters brain structural covariance. Rows and columns of each matrix denote atlas-defined structures, and color intensity the correlation strength (Pearson's correlation, red = positive, blue = negative). (**A**) Mice were kept either in STD or ENR conditions. (**B**) based on patterns of behavior (slope of the RE trajectory), ENR animals were stratified into 'flat' and 'down' roamers. Six mice with positive slopes and 2 intermediates were not included into this visualization. 'Correlatedness' in flat roamers visibly exceeds the ENR (all) pattern, while in 'down roamers' it appears weaker than in STD (see text for the Chi Square statistics of the comparisons).

The online version of this article includes the following figure supplement(s) for figure 3:

**Figure supplement 1.** The violin plots with overlaid data points show increased overall coefficient variance in the volume of mouse brain regions in Flat compared to Down subgroups of ENR-housed mice.

highly correlated with each other, whereas this was not the case in 'down' roamers and STD mice (*Figure 3*). These correlation patterns support the hypothesis of a link between behavioral trajectories and distinct levels of brain structural covariance, which develops independently of genetic variation, or the idea that emerging complex behavioral patterns are dependent on changes in covariance between widely distributed regions.

To test hypotheses regarding specific network connections we used network-based statistics (NBS). To describe the graph model, an appropriate set of nodes was defined using the mouse brain ROIs from the DSURQE atlas that differed significantly in volumes based on the ENR > STD contrast after correction for multiple comparisons (5% FDR; *Figure 2A*, *Supplementary file 1*). Values of bilateral regions were averaged and plotted in both hemispheres. Comparing ENR and STD mice, NBS analysis detected a single statistically significant subnetwork of increased structural covariance, comprising 23 structural connections ($t = 2.4$; $p < 0.05$). By contrast, comparing 'flat' vs. 'down' mice in the ENR group, NBS identified a larger statistically significant subnetwork of increased structural covariance comprising 49 structural connections ($t = 2.4$; $p < 0.05$; *Figure 4A, B*). *Figure 4C, D* depicts how the size of these subnetwork depends on the chosen $t$ thresholds.

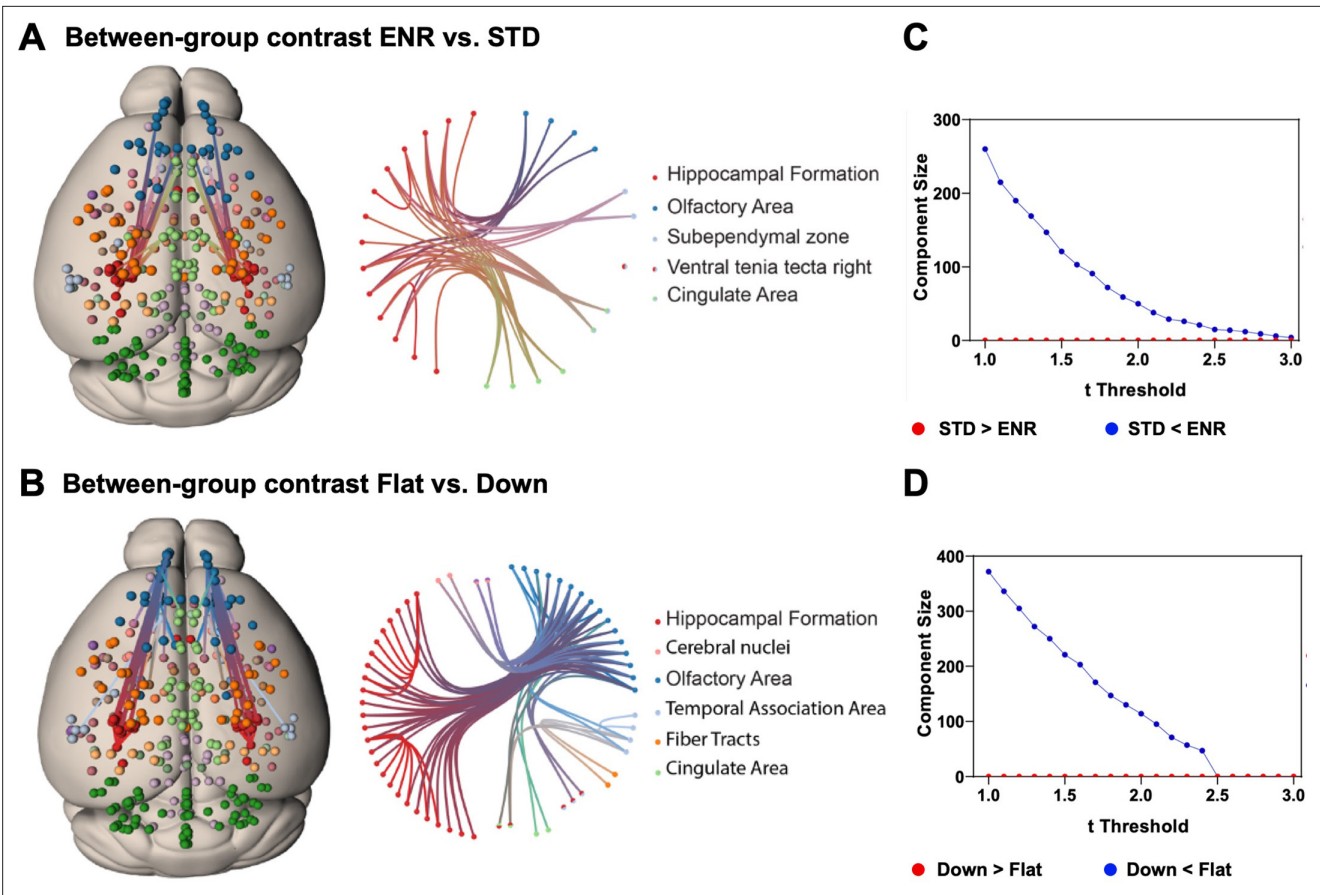

**Figure 4.** Network based statistics confirms differential structural connectivity patterns. Visualization of subnetworks in the contrasts (**A**) ENR > STD and (**B**) FLAT > DOWN both at a threshold of t = 2.4. In the left panels nodes are projected onto a standard mouse brain template (Allen Mouse Common Coordinate Framework v.3). Edges and nodes are colored according to their anatomical location (Dark Green: Cerebellum, Light Green: Cingulate Area, Red: Hippocampal Formation, Orange: Fiber Tracts, Dark Blue: Olfactory Area, Light Blue: Temporal Association Area, Light Pink: Cerebral nuclei, Purple: Cortical Subplate). Nodes were defined using the DSURQE atlas and restricted to the regions of interest with a significant group difference in brain volumes based on the ENR > STD contrast. Values of bilateral regions were averaged and plotted in both hemispheres. Brain connectivity maps and circular connectogram were generated using NeuroMArVL (http://immersive.erc.monash.edu.au/neuromarvl). **C** and **D**: Visualization of the component size of the significant subnetwork depending on the chosen threshold. (**C**) ENR vs STD and (**D**) Flat vs Down. Regions of interest were defined with the DSURQE atlas and restricted to the regions of interest with a significant group difference in brain volumes based on the ENR > STD contrast.

The NBS analysis supports the conclusion that interindividual differences in behavioral trajectories (here clustered into the 'flat' vs. 'down' roamer subgroups of ENR) surpassed the housing effects, and identifies specific subnetworks of structural covariance in the mouse brain.

## Discussion

Correlative analyses of magnetic resonance imaging (MRI) data provide evidence of integrated networks of brain regions, both structurally and functionally. These provide non-invasive insights into the organization of the brain at the macroscale, going beyond the analysis of local changes within individual brain regions as revealed by univariate analysis methods (*Bullmore and Sporns, 2009*). Within this framework, 'structural covariance', permits the study of variation in the organization of brain regional structures within networks that might emerge across a population of individuals in both humans (*Alexander-Bloch et al., 2013*; *Evans, 2013*) and mice (*Bruce et al., 2021*; *Mueller et al., 2021*). Applying structural covariance and NBS analysis to our individuality model, we observed patterns of structural covariance across the mouse brain that are highly distinct between STD- and ENR-housed mice, which may reflect synchronized plastic changes occurring across multiple brain

regions over time. Remarkably, within the ENR group itself, the structural covariance matrix for the 'down' roamers was similar to the matrix generated for standard-housed mice (*Figure 1F, G*). The structural covariance matrix for those mice with a 'flat' RE suggests a much higher degree of inter-regional correlation in comparison to 'down' or STD mice, findings confirmed and extended by the NBS analysis. Variation in mouse brain structural covariance in the ENR group suggests that individual behavioral trajectories (as here first approximated by RE, but in reality encompassing the full scope and complexity of behaviors; *Freund et al., 2015*) are associated with a certain degree of individ-ualization of brain structural networks at the macroscale and that this largely explains differences observed between standard and ENR mice per se.

The two groups of flat vs. down roamers were formed in order to be able to calculate struc-tural covariance matrices at the ends of the behavioral spectrum within the ENR group. We used a within-group effect to point to the existence of interindividual differences, but by necessity, structural covariance has to be calculated in groups and cannot be calculated for individuals with the avail-able structural MR images from this dataset. We excluded two animals with borderline slopes at the boundary between the groups and as a measure of stringency, removed mice with a clearly positive slope from the 'flat' group. Chi-square statistics support the qualitative impression that the visualiza-tion conveys: the patterns differ strongly.

Our data suggest that the behaviors reflected in RE are supported by widely distributed brain circuitry, in particular the hippocampal formation, cingulate and temporal cortices, white matter tracts and olfactory areas. Consistent with this view, neuroanatomical and/or structural covariance changes within these mouse brain circuits have been previously shown to be associated with spatial memory, navigation and cognitive strategies (*Lerch et al., 2011*), social behaviors (*Bruce et al., 2021*), and susceptibility or resilience to stress exposure (*Anacker et al., 2016*), all domains of mouse behavior that may contribute to the individual estimate of RE.

Environmental enrichment has always been by and large a 'black box' paradigm and discussions about the relative contribution of the various factors that make up enrichment (e.g., group size, space, changing objects) have filled volumes, without reaching a unified theory. In the center of our own considerations are learning and social interaction as key components. While the deconstruction of enrichment is an important research avenue, a case can also be made for leaving the black box closed and appreciate enrichment for its ability to apply a complex combination of inanimate and social stimuli. For the present study we have to leave open, to which extent and how exposure to the enriched conditions is causal for the observed behavioral trajectories and the associated changes in structural covariance. Nevertheless, the associations are strongly indicative of at least shared (indirect) causality.

The mice in our study are inbred, thus the genetic background and the 'nominal' environment are identical for all mice in the ENR group. Hence, phenotypic variation at the level of brain and behavior must arise from the non-shared component of the environmental factors ('non-shared environment'). Collectively, our data support a view that a continuum exists between RE and experience-dependent plasticity in brain structure (as measured by ex vivo sMRI). We speculate that exposure to ENR may build on initial variance in mouse brain volumes observable from early in postnatal life (*Qiu et al., 2013*) and amplifies these differences by providing the opportunity for the development of individual behavior (*Freund et al., 2013*; *Kempermann, 2019*; *Körholz et al., 2018*; *Zocher et al., 2020*). Longitudinal in vivo multimodal MRI studies are however required to definitively confirm this view and define the relative contributions of pre-existing individual differences in both functional and structural mouse brain networks as compared to the experience of the shared enriched environment and link these changes to their cellular and molecular correlates.

A certain limitation to our current study is the fact that the stratification of the ENR group into behaviorally distinct subgroups involved arbitrary decisions. With these, however, we intended to maximize the *N* number within each subgroup to increase power, use equally sized groups, and base the definition of the two group on a simple, rational, and plausible parameter (i.e., slope of the behavioral trajectory). The observed differences in covariance are large and occur in a group of mice, which are genetically homogenous and share the same environment. The data are consistent with the hypothesis that greater activity results in greater brain plasticity and, by consequence, a richer connectome. While *Figure 4* offers first insight into which connections are involved, we here primarily emphasize the fact that the number of connections increases with greater activity. Actual effect sizes,

specific network properties, and the role of specific connections will have to be examined in larger follow-up studies based on more and different behavioral measures and stringently objective stratification strategies. As first step into this direction, we however present the continuous correlation between RE slopes and gray matter volumes in *Figure 2B*.

Because 'flat' roamers show stronger structural brain connectivity (based on structural covariance in MRI), it is tempting to speculate that keeping a stable level of territorial coverage has a positive effect on brain networks. In fact, our RE measure has already been used in human studies, providing evidence that greater levels of RE (assessed with a smartphone app) are associated with a greater positive affect and greater hippocampal–striatal functional connectivity (*Heller et al., 2020*). Our animal studies allow the generalization of such observations and ultimately will enable identification of the underlying mechanisms at the cellular and molecular levels.

The by now extensive literature on environmental enrichment covers different overarching mechanistic ideas, including the 'learning theory', the 'arousal theory', and the 'developmental theory'. As argued recently, all of these seem to be correct to a certain extent and contribute to a multifactorial network of causes across scales (*Kempermann, 2019*), RE captures only a small facet of this complexity but appears to be a remarkably strong factor. We hypothesize learning and social factors to make important contributions to the individuality effect. It is, however, an unresolved question, to which extent these enrichment factors are necessary for behavioral and structural individualization to emerge.

Even now, however, our study demonstrates how whole mouse brain structures are distinctively affected by the non-shared component of environmental enrichment and can be linked to stable behavioral trajectories.

## Materials and methods
### Animals and housing conditions
Female C57BL/6JRj mice from as many litters as possible were purchased at 4 weeks of age from Janvier Labs and were randomly divided into standard (STD) ($n$ = 10) and enriched (ENR) ($n$ = 38) housing conditions. ENR mice were subcutaneously injected into their neck with a glass coated micro transponder (SID 102/A/2; Euro I.D.) under brief isoflurane anesthesia. The shared enriched environment took place in a cage system custom-built to our specifications (PhenoSys GmbH, now marketed as 'PhenoSys ColonyRack Individuality 3.0'), consisting of 70 polycarbonate cages (1264C Type II, Tecniplast) that are connected via transparent tunnels and distributed on seven levels. With this cage system, longitudinal tracking of mice is obtained by RFID antennae, located on connecting tunnels (*Kempermann et al., 2022*). For this specific experiment, ENR mice only had access to 35 polycarbonate cages in four levels (total area of 1.37 m²), equipped with toys and hideouts, that were replaced and rearranged once a week. STD mice were housed in two polycarbonate cages (36.5 × 20.7 × 14 cm; Type III, Tecniplast) in groups of five animals per cage. All mice were maintained on a 12-hr light/12-hr dark cycle with 55 ± 10% humidity at the animal facility of the Center for Regenerative Therapies Dresden. Food (#V1534; Sniff) and water were provided ad libitum.

The experiment was conducted in accordance with the applicable European and national regulations and approved by the local authority (Landesdirektion Sachsen, file number 7/2016 TVT DD24 5131-365-8-SAC). All analyses were performed in a blinded manner.

### Analysis of the tracking data
Antenna contacts, resulting from mice activity, were recorded with the software PhenoSoft Control (PhenoSys GmbH), which identifies the specific antenna and mouse, as well as the time stamp of the antenna contact. As previously described (*Kempermann et al., 2022*), Shannon entropy of the roaming distribution was calculated as

$$\mathrm{RE}_{i,t} = - \sum_{j=1}^{k} \left( p_{i,j,t} \log p_{i,j,t} \right) / \log \left( k \right)$$

where $i$ is the mouse, $j$ is the antenna, $k$ is the total number of antennae, and $t$ is the day. RE quantifies differences in territorial coverage, and because mice are nocturnal animals, only the dark phase

of the cycle was used for this analysis. From the nightly mean RE values, four time blocks (T1, T2, T3, and T4), of 21 calendar days each, was generated.

## Mixed linear models and repeatability estimation

To investigate whether the observed phenotypic variances are resulting from inter- and/or within-individual variability, we employed generalized linear mixed models in a Bayesian framework, as previously described (*Fong et al., 2010*; *Zocher et al., 2020*). Briefly, repeatability (*R*) is the fraction of total variance that can be attributed to interindividual differences, rather than within individual differences, calculated as $R = \frac{V_{(ind)}}{V_{(ind)} + V_{(res)}}$ , where $V_{(ind)}$ represents the interindividual variances and $V_{(res)}$ the residual, or within individual, variances (*Dingemanse and Dochtermann, 2013*). The behavioral phenotypes from RE were mean centered and scaled to unity variance. We used time blocks as fixed effect and an interaction between time block and individual identifier as random effect. With a random effect, it was possible to estimate interindividual variances for each time block, while the residual variances were estimated separately. To fit the model, we applied Markov chain Monte Carlo estimation with Gibbs sampling (MCMCglmm R package). A Gaussian error distribution was assumed with weakly informative default priors for the fixed effect (time blocks). An inverse Wishart distribution prior was selected for residual variances. From the posterior (co)variance distributions, we derived the estimates of repeatability and interindividual correlations, using a mode of the posterior density.

## Within-enrichment subgroups based on slope patterns

The optimal number of clusters was determined by a silhouette analysis using an unsupervised partitioning method of clustering (*k*-means), which measures the quality of clustering for each data point (*Kaufman and Rousseeuw, 1990*). All enriched mice were clustered based on mean RE values for each time period (T1, T2, T3, and T4).

A visual assessment of individual behavioral trajectories, based on RE, indicates two distinct patterns: fixed amount of territorial coverage ('flat') or behavioral habituation over time ('down'). After detecting an optimal number of two clusters within our enriched group, we applied slope values, calculated as $b = \frac{\sum \left(x - \bar{x}\right)\left(y - \bar{y}\right)}{\sum \left(x - \bar{x}\right)^2}$ , where *x* are time points (1, 2, 3, and 4, representing time blocks), *y* are RE values for each time point, and $\bar{x}$ and $\bar{y}$ are sample means of the known *x*'s and the known *y*'s. The slope values fall into a small distance range: −0.049 up to 0.016. We decided to select up to 15 animals for each subgroup, due to the uncertain behavioral pattern that some mice showed. Finally, 15 animals from the lower spectrum of slope values (−0.049 to −0.008) were grouped as 'down' roamers, and 15 animals with slope values ranging from −0.003 up to 0.004 were classified as 'flat' roamers. Excluded from the subgroup analysis are two mice with in-between slope values and six mice with higher than 0.006 slope values.

## Tissue preparation for ex vivo MRI

All animals were killed at 17 weeks of age by cardiac perfusion performed on site at the Kempermann Lab, Center for Regenerative Therapies Dresden, prior to shipping to King's College London (KCL). The details of the perfusion protocol have been published elsewhere (*Richetto et al., 2017*; *Wood et al., 2016*). Briefly, mice were anesthetized with sodium pentobarbital (Narcoren 16 g/100 ml; 5 µl/g; i.p.) and intracardially perfused with 30 ml of 0.1 M phosphate-buffered saline (PBS), containing 10 U/ml heparin, followed by 30 ml of 4% (vol/vol) paraformaldehyde (PFA). Post-perfusion, the mouse was decapitated, and the skin, ears, and lower jaw removed, and the brain left within the cranium. This is done to minimize deformations due to dissection. Each brain is then first incubated for 24 hr in 4% PFA solution and then placed in 0.1 M PBS containing 0.05% (wt/vol) sodium azide. Samples were then shipped to KCL and stored at 4°C in this solution for 4 weeks prior to ex vivo MR image acquisition to allow tissue rehydration. MRI examiners were blinded for the experimental groups.

## Tissue preparation for immunohistochemistry

Brains in the skulls were shipped back to Dresden from the KCL. At the German Center for Neurodegenerative Diseases (DZNE) Dresden, brains were dissected from the skulls, incubated in 30% sucrose in phosphate buffer for 2 days and cut into 40-µm coronal sections using a dry-ice-cooled copper

block on a sliding microtome (Leica, SM2000R). Sections were stored at 4°C in cryoprotectant solution (25% ethylene glycol, 25% glycerol in 0.1 M phosphate buffer, pH 7.4). For detection of DCX-positive cells, the peroxidase method was applied. Free-floating sections were pretreated with 0.2 M boric acid (pH 9) at 70°C for 1 hr as an antigen retrieval method, washed in 1% phosphate buffer (PBS) and then incubated for 12 hr at 4°C in PBS with 10% donkey serum (Jackson Immuno Research Labs) and 0.2% Triton X-100 (Carl Roth). After the protein blocking step, sections were incubated overnight at 4°C with the primary antibody (goat anti-DCX, 1:250; Santa Cruz Biotechnology, Cat# sc-50548, RRID:AB_2079663) diluted in PBS containing 3% donkey serum and 0.2% Triton X-100. Following washing steps, incubation with biotinylated secondary antibody (1:500, Jackson Immuno Research Labs Cat# 705-065-147, RRID:AB_2340397) occurred for 3 hr at room temperature. Sections were then incubated in 0.6% hydrogen peroxide in PBS for 30 min to inhibit endogenous peroxidase activity. Detection was performed using the Vectastain ABC Elite reagent (9 μg/ml of each component, Vector Laboratories, LINARIS) with diaminobenzidine (0.075 mg/ml; Sigma) and 0.04% nickel chloride as a chromogen. Stained sections were mounted onto glass slides, cleared with Neo-Clear (Millipore), and cover-slipped using Neo-Mount (Millipore). DCX-positive cells were counted on every sixth section along the entire rostrocaudal axis of the dentate gyrus using a brightfield microscope (Leica DM 750). Experimenters were blinded for the experimental groups.

## MR image acquisition

A 9.4T Bruker BioSpec 94/20 horizontal small bore magnet (Bruker Ltd, UK) and a quadrature volume radio-frequency coil (39-mm internal diameter, Rapid Biomedical GmbH, GER) were used for all ex vivo sMRI acquisitions. Fixed brain samples were placed securely, four at a time, in a custom-made MR-compatible holder and immersed in proton-free susceptibility matching fluid (Fomblin; Solvay, UK). Samples were scanned in a random order, with the operator blinded to sMRI acquisitions. Fixed brain samples were placed securely, four at a time, in a custom-made MR-compatible holder and immersed in proton-free susceptibility matching fluid (Fomblin; Solvay, UK). Samples were scanned in a random order, with the operator blinded to treatment group by numerical coding of samples. Scanning was interspersed with phantoms to ensure consistent operation of the scanner. T2-weighted images were acquired using a 3D fast spin-echo (FSE) sequence with the following parameters: effective echo time (TE) 30 ms, repetition time (TR) 3000 ms, field of view 25 × 25 × 20 mm and acquisition matrix 250 × 250 × 200 yielding isotropic voxels of 100 μm$^3$, scan time = 5 hr and 44 min. This MRI resolution (100 μm$^3$) is at the lower end for ex vivo anatomy as compared to other studies, which employ longer acquisition times with specific pulse sequences and the use of contrast enhancement, for example, the inclusion of gadolinium in the perfusion and rehydration steps to reach resolutions as low as 30–40 μm$^3$ (*Bruce et al., 2021*). We adopted the protocol reported for this study based on our prior work (e.g., *Lin et al., 2021 Mueller et al., 2021*) as a pragmatic initial step, which allows us to collect both structural and diffusion tensor imaging data (the latter to be reported elsewhere) in a single overnight scan across multiple mice. Of note however, prior work comparing low- and high-resolution ex vivo sMRI scans on the same population of mice suggests that the majority of variance is explained by biology and not methodology (*Lerch et al., 2012*). Nonetheless, sMRI studies at much higher resolution using contrast enhancement are now underway to address this issue directly.

## MR image processing

MR images were visually inspected in native space for artefacts, with no images excluded on this basis. Raw MR images were converted from the manufacturer's proprietary format to the NIFTI format and processed using a combination of FSL (*Jenkinson et al., 2012*), ANTs (*Avants et al., 2011*), and the Quantitative Imaging Tools (QUIT) package written in C++ software utilizing the ITK library, available from https://github.com/spinicist/QUIT. The following steps were performed on the T2-FSE anatomical MR images in their native space. A Tukey filter was applied to the FSE MR images in *k*-space to remove high-frequency noise followed by correction for intensity inhomogeneity using the N4 algorithm (*Tustison et al., 2010*). A study-specific template was then constructed from MR images of *n* = 24 mice randomly selected from the entire dataset, using the *antsMultivariateTemplateConstruction2. sh* script with cross-correlation metric and SyN transform (*Avants et al., 2010*). The template and individual brains were skull stripped using the RATS algorithm implemented in QUIT using the *qimask* script (*Wood et al., 2016*). All MRI data can be found here: https://osf.io/m7gpd/.

## MR image analyses

### ABS of regional brain volumes

Group-level differences in volume were assessed using a combination of ABS and voxel-wise DBM as in our prior work (*Mueller et al., 2021*). To enable ABS analysis of regional brain volumes, the study-specific template was then registered to the Dorr-Steadman-Ullmann-Richards-Qiu-Egan (DSURQE) mouse brain atlas (40 µm) (https://wiki.mouseimaging.ca/display/MICePub/Mouse+Brain+Atlases). This atlas has 182 individual mouse brain structures defined (including left and right labels for most structures, giving 356 labels in total). The T2-weighted 3D FSE images for all study subjects were then non-linearly registered to the study template using the *antsRegistrationSyN.sh* script. For the DSURQE atlas, the inverse transforms from the atlas to the study template and from the study template to each subject, were applied to calculate the brain and atlas-based ROI volumes for each subject (*Doostdar et al., 2019*; *Kuan et al., 2021*). After careful checking of atlas label alignment to each individual subject's MR images, we automatically extracted volumes for the 182 ROIs comprising the DSURQE atlas, merging the left and right labels for each ROI. Total brain volume was calculated from the summation of the individual atlas ROI volumes. Both absolute volumes ($mm^3$) and relative volumes were compared to assess brain scaling. Relative volumes were calculated as a percentage of total brain volume that is ([(brain region volume)/(whole-brain volume) × 100]). Whilst others and we have used this approach there are caveats. Specifically, whilst some brain regions scale linearly to total brain volume, such as the hippocampus, other regions, such as the cerebellum, may not (*Mankiw et al., 2017*). Hence, it is important to compare both absolute and relative volumes in assessing group differences. Group-level differences in either absolute or relative volumes were compared between STD and ENR mice using multiple *t*-tests (two-tailed, unequal variance assumed) with $\alpha = 0.05$ using R-project (v4.0). The resulting p values were subsequently corrected for multiple comparisons using the FDR at 5% ($q < 0.05$) using Prism software (v8.4.2; GraphPad, La Jolla, CA, USA). To calculate the magnitude and direction of volume change for each region between groups, effect sizes for each brain region were calculated using the standardized mean difference (effect size = $(\mu_{[ENR]} - \mu_{[STD]}/\sigma_{STD})$); measured in units of standard deviation. The volumetric MRI data are found in *Supplementary file 1* (absolute values) and 2 (relative values).

## Deformation-based morphometry

Jacobian determinant maps were calculated from the inverse warp fields in standard space using the *CreateJacobianDeterminantImage* script (ANTs) and log-scaled to allow voxel-wise estimation of apparent volume change via deformation-based morphometry (DBM) (*Mueller et al., 2021*). To compare local volume differences between STD and ENR mice, voxel-based nonparametric statistics were performed on the log-transformed Jacobian determinant maps using FSL randomize as previously described (*Mueller et al., 2021*). The resulting *F*-statistic maps were corrected for multiple comparisons using the FWE rate ($p < 0.05$). Data in the manuscript are also shown at $p < 0.05$ uncorrected for multiple comparisons. DBM analyses were run with and without total brain volume as a covariate of no interest, to check for potential global scaling effects.

## Correlation between RE and local volume changes

The slope of the RE metric, calculated across all time blocks was regressed against log-Jacobians of each individual at each voxel in the ENR cohort only. The resultant correlation map was then thresholded using the FWE rate ($p < 0.05$). Data in the manuscript are also shown at $p < 0.05$ uncorrected for multiple comparisons.

## Mouse brain structural covariance

Following the atlas-based approach, we explored patterns of structural covariance using relative brain volumes from 182 predefined structures in the DSURQE atlas (*Yee et al., 2018*). We computed the Pearson correlation coefficient between all relative structure volumes, resulting in a 182 × 182 matrix of correlations representing the structural covariance network, for both standard housing and enriched housing groups. We repeated the same analysis considering the within-enrichment subgroups 'flat' and 'down' roamers. The 182 structures are ordered following hierarchical clustering of the enriched-housed correlation matrix, with the same order applied for all matrices.

## NBS analysis

NBS is a powerful statistical method for identifying a statistically significant cluster of connections indicating differences between groups on intermediate network scales (*Zalesky et al., 2010*). As described in previous functional studies (*Ehrlich et al., 2015*; *Geisler et al., 2020*), NBS is computed using the following steps: (1) identify all connections (pairs of nodes) that are different between groups beyond a particular *t*-value (called primary threshold), (2) select the largest contiguous cluster of these connections, and (3) validate the cluster's significance by permutation testing. In permutation testing an empirical null distribution of the largest cluster size is generated by conducting the first two NBS steps on resampled group membership data 10,000 times. The returned subnetwork is statistically significant at an FWE corrected value of p < 0.05.

Although the network needs to be considered as a whole, the extent of the returned network can be varied using a different primary threshold. This adjusts the extremity of deviation in a connection between groups required, before it is considered for inclusion in the NBS result. NBS returns a single p value, which represents the likelihood that the subnetwork (also called component) is as observed if in fact there is no difference (i.e., the null hypothesis is true). This approach measures the entire cluster of returned connections but does not identify the contribution of each connection independently. The NBS procedure was carried out for the correlation matrices with a primary threshold of *t* = 3.1 (which corresponds to p = 0.001). The graphic visualization in *Figure 4* is at *t* = 2.4.

As part of an additional analysis to see if cortical regions that are susceptible to thickness reductions are more interconnected the NBS procedure was carried out for STD and ENR on a subgroup of ROIs consisting of all significantly changed regions between STD and ENR.

## Acknowledgements

Helmholtz Association (GK, ANS); Technische Universität Dresden (GK, ANS); Coordination for the Improvement of Higher Education Personnel (CAPES), Brazil (DOC Pleno / Processo nr. 88881.129646/2016-01) (JBL); The Joachim Herz Foundation (JBL); Medical Research Council (New Investigator Research Grant MR/N025377/1 (AV); Centre Grant MR/N026063/1) (AV); TransCampus (TU Dresden and King's College London) research award (AV, GK); DFG research grant: EH 367/7-1 '*Dynamische Veränderungen des strukturellen und funktionellen Hirn-Konnektoms bei Patientinnen mit Anorexia Nervosa*' (SE).

## Additional information

### Funding

| Funder | Grant reference number | Author |
| --- | --- | --- |
| Helmholtz Association | Basic Funding | Anna N Senko<br>Gerd Kempermann |
| Medizinische Fakultät Carl Gustav Carus, Technische Universität Dresden | Basic Funding | Anna N Senko<br>Gerd Kempermann |
| Coordenação de Aperfeiçoamento de Pessoal de Nível Superior | 88881.129646/2016-01 | Jadna Bogado Lopes |
| Joachim Herz Stiftung | | Jadna Bogado Lopes |
| Medical Research Council | New Investigator Research Grant MR/N025377/1 (AV) | Anthony C Vernon |
| TransCampus | TransCampus Research Award | Anthony C Vernon<br>Gerd Kempermann |
| Deutsche Forschungsgemeinschaft | EH 367/7-1 | Stefan Ehrlich |
| Medical Research Council | Centre Grant MR/ N026063/1 | Anthony C Vernon |

| Funder | Grant reference number | Author |
| --- | --- | --- |

The funders had no role in study design, data collection, and interpretation, or the decision to submit the work for publication.

## Author contributions

Jadna Bogado Lopes, Conceptualization, Investigation, Visualization, Methodology, Writing - original draft, Project administration, Writing – review and editing; Anna N Senko, Data curation, Formal analysis, Methodology, Writing – review and editing; Klaas Bahnsen, Formal analysis, Visualization, Writing – review and editing; Daniel Geisler, Investigation, Visualization, Writing – review and editing; Eugene Kim, Investigation, Writing – review and editing; Michel Bernanos, Investigation, Visualization, Methodology, Writing – review and editing; Diana Cash, Formal analysis, Investigation, Writing – review and editing; Stefan Ehrlich, Formal analysis, Investigation, Visualization, Writing – review and editing; Anthony C Vernon, Conceptualization, Formal analysis, Funding acquisition, Visualization, Methodology, Writing - original draft, Writing – review and editing; Gerd Kempermann, Conceptualization, Supervision, Funding acquisition, Visualization, Methodology, Writing - original draft, Project administration, Writing – review and editing

## Author ORCIDs

Anna N Senko (b) http://orcid.org/0000-0003-0885-0440
Klaas Bahnsen (b) http://orcid.org/0000-0002-5413-0359
Eugene Kim (b) http://orcid.org/0000-0003-0066-7051
Stefan Ehrlich (b) http://orcid.org/0000-0003-2132-4445
Anthony C Vernon (b) http://orcid.org/0000-0001-7305-1069
Gerd Kempermann (b) http://orcid.org/0000-0002-5304-4061

## Ethics

The experiment was conducted in accordance with the applicable European and national regulations and approved by the local authority (Landesdirektion Sachsen, file number 7/2016 TVT DD24 5131-365-8-SAC). All analyses were performed in a blinded manner.

## Decision letter and Author response

Decision letter https://doi.org/10.7554/eLife.80379.sa1
Author response https://doi.org/10.7554/eLife.80379.sa2

# Additional files

## Supplementary files

• Supplementary file 1. Absolute volumes.

• Supplementary file 2. Relative volumes.

• MDAR checklist

## Data availability

The structural MR images used in this manuscript are publicly available on the OSF platform (https://osf.io/m7gpd/). The volumetric MRI data are found in Supplementary Files 1 (absolute values) and 2 (relative values). The behavioral data from the cage (animal IDs with time-stamped raw antenna contacts) are assessible at Dryad: https://doi.org/10.5061/dryad.bzkh189ds.

The following datasets were generated:

| Author(s) | Year | Dataset title | Dataset URL | Database and Identifier |
| --- | --- | --- | --- | --- |
| Bogado Lopes J, Senko AN, Bahnsen K, Geisler D, Kim E, Bernanos M, Cash D, Ehrlich S, Vernon AC, Kempermann G | 2022 | Individual behavioral trajectories shape whole-brain connectivity in mice: MRI data | https://osf.io/m7gpd/ | Open Science Framework, m7gpd |

*Continued on next page*

*Continued*

| Author(s) | Year | Dataset title | Dataset URL | Database and Identifier |
|---|---|---|---|---|
| Kempermann G, Lopes J, Senko A, Bahnsen K, Geisler D, Kim E, Bernanos M, Cash D, Ehrlich S, Vernon A, Kempermann G | 2023 | Individual behavioral trajectories shape whole-brain connectivity in mice | https://dx.doi.org/10.5061/dryad.bzkh189ds | Dryad Digital Repository, 10.5061/dryad.bzkh189ds |

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
