## [Editor Report]

This important work is of broad interest to readers studying brain plasticity, individuality, and shared/non-shared environments. The identification of distinct patters of brain networks, in the absence of main effects, between two broad classes defining how mice explore their environments, is especially interesting. The evidence supporting their conclusions is convincing.

---

## [Decision Letter]

**Decision letter after peer review:**

Thank you for submitting your article "Individual behavioral trajectories shape whole-brain connectivity in mice" for consideration by *eLife*. Your article has been reviewed by 3 peer reviewers, including Jason Lerch as Reviewing Editor and Reviewer #1, and the evaluation has been overseen by Timothy Behrens as the Senior Editor. The following individual involved in review of your submission has agreed to reveal their identity: Yohan Yee (Reviewer #2).

Essential revisions:

1) The division of their cohort into two groups raised concerns by all three reviewers. The authors need to address these concerns by clearly showing the existence of two distinct groups, exploring sensitivity to different cutoffs, better justifying why some mice were excluded entirely, and if an intermediate group of mice are still excluded describing their covariance patterns.

2) The manuscript would also benefit from directly using their key behaviour metric, roaming entropy (RE), by using RE-slopes as continuous variables (in addition to clustering based on RE-slopes).

3) The authors need to slightly expand their statistical analyses by running permutation tests over the chi-squared tests, exploring a range of thresholds for network based statistics, and assessing whether alternate means of removing overall brain volume change their results.

4) The authors need to partially expand their introduction and discussion by being clearer that structural covariance, as used here, is a group measure and not easily applicable to an individual, being more nuanced in associating structural covariance with connectivity, and discussing what aspects of their environment (social vs space) have impacts on structural covariance.

Also, a note of clarification based on discussions with reviewer #2: "On acquiring more data to test whether their elaborate enrichment setup is _required_ for individuality, I don't think that's necessary, since that's not specifically the aim of this study. I should have been clearer about my comment – I meant that more as a Gedanken experiment to illustrate what would be needed to specifically tie their enrichment setup to the increasing variability in roaming entropy."

*Reviewer #1 (Recommendations for the authors):*

Most of my confusion and possibly critique relates to the division of mice into two groups. Here the authors need to show the data that supports this division in a more convincing fashion, clear up how the automated clustering supports these division, illustrate how the intermediate group behaves on their measures, and possibly show sensitivity analyses as to how sensitive the groupings are to differences in ways of subdividing the cohort. In addition to clearing up the clustering it would be good to show slopes as a distribution/histogram to be convince the reader that it is indeed a bimodal distribution.

*Reviewer #2 (Recommendations for the authors):*

I genuinely enjoyed reading this paper-it was well written, clear on what was done, concise, and on a relevant and interesting topic.

A common question that came up throughout my reading of the manuscript was on choices made in grouping data: why partition mice into flat and down roamers (while excluding a few with slopes above and in between), as opposed to directly working with the slope. For example, in looking at structural covariance differences, a single analysis (on the entire group of mice) examining covariance differences due to RE-slope could be achieved through the inclusion of RE-slope interaction terms in linear models. Such an approach would benefit from a larger sample size and increased power. Unless the distribution of RE-slopes is bimodal, the justification for stratification is unclear. Similarly, why average the RE over three weeks? Given the temporal resolution of the RFID data and that RE is computed for each night, partitioning into four long periods seems unnecessary.

Clarity on the role of the enriched environment would be helpful. While it is reasonable to believe that individuality emerges from exposure to an enriched environment, it is unclear from this study if this enrichment is *required*. An explicit comparison of enriched mice with those raised in standard housing would rule out intrinsic age-associated divergence of roaming entropy. In other words, if four cohorts of standard-housed mice were exposed to the enriched environment to track roaming but only during each of the four time periods, would the emergence of individuality not be as pronounced?

Some comments on specific parts of the study and manuscript:

1) Line 100: "Mice that showed in-between slope values (n = 10) were excluded from the clustering" is inconsistent with the subject numbers listed on lines 88 (total n=38) and 96-97 (15 flat roamers, 15 down roamers). Also, this conflicts with the statement on lines 393-394 ("Excluded from the subgroup analysis are 2 mice with in-between slope values and 6 mice with higher than 0.006 slope values").

2) Line 165: do you mean Figure 2B (as opposed to 3B)?

3) Line 175: I would suggest averaging over cortical and subcortical regions separately, given their distinctive function, development, gene expression, connectivity, and covariance patterns.

4) Line 282: missing closing parenthesis.

5) There are various places in the manuscript where structural covariance is used as a proxy to traditional measures of connectivity. While structural covariance has been found to be statistically associated structural and functional connectivity, the idea that structural covariance is driven by synchronized plasticity between structurally connected regions has yet to be confirmed. Therefore, I would caution against making this connection (no pun intended) between covariance and connectivity. A mention of this (and other study limitations) in the discussion might be helpful.

6) Lines 550-551: Could you double check the definition of the p-value? Unless NBS (which I am admittedly not an expert on) defines the p-value differently, the p-value is a probability of observing data given the null, as opposed to a probability of an effect.

7) Reasons for only including female mice should be explicitly listed – I assume male mice were excluded to avoid aggression/mating?

8) Figure 3: it would be helpful to annotate the rows and columns of these matrices with names of coarse structures.

Potential ideas to further improve the study include:

1) Are anatomical changes driven by neurogenesis? I.e., does the size of the dentate gyrus correlate with DCX?

2) Apart from access to a larger and richer space, the ENR group differs from STD in the number of mice and therefore potential social interactions. I'd be curious to know whether these results can be separated into social vs spatial exploration effects. For example, based on RFID data, do flat roamers tend to spend time with other mice more, while down roamers are more isolated? If it is possible to extract from the RFID data, I think an analysis that includes measures of social interactions would make this a far stronger study.

3) Similar to the above comment, can the effect of activity be teased out from the RFID data? Exercise has been shown to correlate with brain structural differences; I wonder if some of the neuroanatomical variation seen comes about from mice just moving around more.

*Reviewer #3 (Recommendations for the authors):*

– The interpretation of structural covariance is oversimplified especially in introduction. Specifically, REF 19 is being mischaracterized – this paper did suggest some convergence but also significance divergence with white matter connectivity. In the discussion the authors invoke the idea that structural covariance "may reflect synchronized plastic changes occurring across multiple brain regions over time". This idea should be included in the motivation for the investigation as well.

– Given references to prior related work by the authors and others, it is not currently clear what are the exact analyses that have not previously been done which are unique to this paper. The paper would benefit from a clear discussion paragraph stating what this study does that has not been done previously and what is a replication (differentiating replication from restatement of results already reported in the same mice).

– The statement on p.7 that "The majority of these ROIs were conserved as significant group differences when considering relative volumes (% of whole brain), suggesting normal scaling" should be clarified. Do the authors mean normal scaling versus nonlinear scaling cf https://www.science.org/doi/10.1126/science.aar2578? Note that dividing by total brain size is not really an effective way to "control" for the effect of total brain size (https://www.sciencedirect.com/science/article/pii/S1053811922006012)

– The chi-square tests for equality of two correlation matrices would be better if replicated with permutation testing as REF 21 seems to indicate these tests are anti-conservative in small samples.

– NBS uses permutation testing which is good. However it is more rigorous to consider the t-statistic over a range of thresholds not just a single threshold. I'd suggest 2.5-3.5 in 0.1 intervals as a sensitivity analysis. Also it's not clear why t=2.4 is mentioned in results but t=3.1 in mentioned in the methods.

– This is out of my area of expertise but is it possible that genetic differences including de novo mutations could underlie some inter-individual differences between inbred mice?

– I am having trouble understanding the rationale for the precise cut off values between groups. On p.15 manuscript says "Excluded from the subgroup analysis are 2 mice with in-between slope values and 6 mice with higher than 0.006 slope values." while on p.10 it says "Mice that showed in-between slope values (n = 10) were excluded from the clustering to achieve a sharper distinction. Presumably, I am not understanding something but it should be more clear.

– In addition, the rationale for excluding "up roamers" with higher slope values is not clear. These mice should be included in the flat group in a sensitivity analysis to assess robustness of findings.

– Currently it is not clear to what extent findings depend on the specific thresholds chosen to differentiate flat vs down roamers. It would be an improvement to conduct a sensitivity analysis in order to confirm the robustness of the findings to methodological variability to methodological choices which may seem arbitrary to a reader.

---

## [Author Response]

Essential revisions:1) The division of their cohort into two groups raised concerns by all three reviewers. The authors need to address these concerns by clearly showing the existence of two distinct groups, exploring sensitivity to different cutoffs, better justifying why some mice were excluded entirely, and if an intermediate group of mice are still excluded describing their covariance patterns.

The division into flat vs. down roamers reflects the extremes of a continuous spectrum of associations between the Roaming Entropy (RE) trajectories and gray matter volume changes. In order to describe differences in structural covariance patterns however, groups have to be formed within this continuum; for individual mice no covariance exists. For “individualised“ matrices one would need to acquire multiple different sequences and use a morphometric similarity mapping approach (Seidlitz et al., Neuron, 2018), which was not (yet) done here. We describe that mice that are at the upper end of the RE activity spectrum show a greater degree of “*correlatedness*” of gray matter volume changes than those at the lower end. This difference in covariance patterns is visualized in the correlation matrices of Figure 3. By necessity the borders of these extreme groups are arbitrary. If the N in each group is too small, the matrix is less robust than with greater N. Splitting the entire group of animals in the Individuality cage in two, comparing the lower and the upper half yields essentially the same result. Omitting the mice with upward slopes and two intermediates increases the sharpness of the distinction in the visualization.

Note that we take the patterns as patterns only and do not interpret individual pairs of correlating areas. To the extent such approach was reasonable in the context of the given study, this was achieved with the graph theory analysis presented in Figure 4. The conclusion from Figure 3, in contrast is, that the structural covariance patterns are different between mice at the upper and the lower end of the scope of RE trajectories within the same environment. We point to the variance in structural covariance that is induced by enrichment.

“The two groups of flat vs. down roamers were formed in order to be able to calculate structural covariance matrices at the ends of the behavioral spectrum within the ENR group. We used a within-group group-effect to point to the existence of inter-individual differences, but by necessity, structural covariance has to be calculated in groups and cannot be calculated for individuals with the available structural MR images from this dataset. We excluded two animals with borderline slopes at the boundary between the groups and as a measure of stringency, removed mice with a clearly positive slope from the ‘flat’ group. Chi Square statistics support the qualitative impression that the visualization conveys: the patterns differ strongly.”

2) The manuscript would also benefit from directly using their key behaviour metric, roaming entropy (RE), by using RE-slopes as continuous variables (in addition to clustering based on RE-slopes).

This has been done and is now presented as new Figure 2B which clearly shows positive correlations that survive multiple comparisons correction (FWE p<0.05) between RE slope and local volume changes. We agree that this is a helpful additional piece of information. We have retained the correlation between the RE slope and the down / flat sub-groups as a new supplementary figure, noting that this largely recapitulates the pattern seen in the revised Figure 2B across all mice, but with no statistically significant changes.

3) The authors need to slightly expand their statistical analyses by running permutation tests over the chi-squared tests, exploring a range of thresholds for network based statistics, and assessing whether alternate means of removing overall brain volume change their results.

We now present the NBS results for both contrasts across a range of common t-thresholds (see added panels C and D to Figure 4). The clusters’ significance was validated by permutation testing. In line with the main results (shown at our a priori threshold of t=2.4 (ca. p=0.008, i.e. p<0.01)) the figure shows that similar components are present at more lenient thresholds (as expected those components are somewhat larger) and also more conservative thresholds (components grow smaller, as expected). Overall these data speak to the robustness of the NBS results.

4) The authors need to partially expand their introduction and discussion by being clearer that structural covariance, as used here, is a group measure and not easily applicable to an individual, being more nuanced in associating structural covariance with connectivity, and discussing what aspects of their environment (social vs space) have impacts on structural covariance.

We appreciate this comment, as these are critically important points that no reader should miss. We have carefully added clarifications throughout the manuscript.

“Open questions about the exact nature of the connectome and how it is represented in covariance patterns as assessed by imaging studies abound, especially with respect to the crucial issue of causes and consequences.”

“Environmental enrichment has always been by and large a “black box” paradigm and discussions about the relative contribution of the various factors that make up enrichment (e.g. group size, space, changing objects have filled volumes), without reaching a unified theory. In the center of our own considerations are learning and social interaction as key components. While the deconstruction of enrichment is an important research avenue, a case can also be made for leaving the black box closed and appreciate enrichment for its ability to apply a complex combination of inanimate and social stimuli. For the present study we have to leave open, to which extent and how exposure to the enriched conditions is causal for the observed behavioral trajectories and the associated changes in structural covariance. Nevertheless, the associations are strongly indicative of at least shared (indirect) causality.”

Also, a note of clarification based on discussions with reviewer #2: "On acquiring more data to test whether their elaborate enrichment setup is _required_ for individuality, I don't think that's necessary, since that's not specifically the aim of this study. I should have been clearer about my comment – I meant that more as a Gedanken experiment to illustrate what would be needed to specifically tie their enrichment setup to the increasing variability in roaming entropy."

We agree that addressing this question experimentally would go far beyond the scope of the present study. But the point is very interesting and we have added a short statement in this direction to the end of the discussion. These are in fact thoughts that guide the future direction of our work (See quote in the previous response).

Reviewer #1 (Recommendations for the authors):Most of my confusion and possibly critique relates to the division of mice into two groups. Here the authors need to show the data that supports this division in a more convincing fashion, clear up how the automated clustering supports these division, illustrate how the intermediate group behaves on their measures, and possibly show sensitivity analyses as to how sensitive the groupings are to differences in ways of subdividing the cohort. In addition to clearing up the clustering it would be good to show slopes as a distribution/histogram to be convince the reader that it is indeed a bimodal distribution.

The clustering was not used for the presented analysis. The appearance of the related sentences in the method section was an oversight that has now been corrected.

Reviewer #2 (Recommendations for the authors):I genuinely enjoyed reading this paper-it was well written, clear on what was done, concise, and on a relevant and interesting topic.A common question that came up throughout my reading of the manuscript was on choices made in grouping data: why partition mice into flat and down roamers (while excluding a few with slopes above and in between), as opposed to directly working with the slope.

We did this and present the data now. See our comment to Referee #1. The stratification allowed the visualization of the differences in the covariance matrices.

For example, in looking at structural covariance differences, a single analysis (on the entire group of mice) examining covariance differences due to RE-slope could be achieved through the inclusion of RE-slope interaction terms in linear models. Such an approach would benefit from a larger sample size and increased power. Unless the distribution of RE-slopes is bimodal, the justification for stratification is unclear.

Using the extremes allows a clearer visualization. We are not interpreting any particular correlation between regions but look at the overall differences in “correlatedness”.

Similarly, why average the RE over three weeks? Given the temporal resolution of the RFID data and that RE is computed for each night, partitioning into four long periods seems unnecessary.

This is done for clarity and consistency with the other studies we have done (cf. Körholz et al. 2018 and Zocher et al. 2020). We are currently working on a new analysis pipeline that resolves this issue.

Clarity on the role of the enriched environment would be helpful. While it is reasonable to believe that individuality emerges from exposure to an enriched environment, it is unclear from this study if this enrichment is required. An explicit comparison of enriched mice with those raised in standard housing would rule out intrinsic age-associated divergence of roaming entropy. In other words, if four cohorts of standard-housed mice were exposed to the enriched environment to track roaming but only during each of the four time periods, would the emergence of individuality not be as pronounced?

This is a very interesting and important question, but it actually comprises many sub-questions, some of which are very difficult to address and go well beyond the scope of the present study. The enrichment literature is full of potential explanations and speculation about the most active ingredients. In our current study we have left the black box closed, but believe that learning and social contexts are important drivers. It is absolutely correct, however, that stochastic effects and what the reviewer calls “intrinsic age-associated divergence” are important. The latter is a challenging idea, however: could such divergence indeed be genetically determined but become differentially apparent, despite the fact that all mice share the same genotype? We have added a short paragraph to the discussion to point to this interesting speculation.

“The by now extensive literature on environmental enrichment covers different overarching mechanistic ideas, including the “learning theory”, the “arousal theory” and the “developmental theory”. As argued recently, all of these seem to be correct to a certain extent and contribute to a multi-factorial network of causes across scales (Kempermann 2019), RE captures only a small facet of this complexity but appears to be a remarkably strong factor. We hypothesize learning and social factors to make important contributions to the individuality effect. It is, however, an unresolved question, to which extent these enrichment factors are necessary for behavioral and structural individualization to emerge.”

Some comments on specific parts of the study and manuscript:1) Line 100: "Mice that showed in-between slope values (n = 10) were excluded from the clustering" is inconsistent with the subject numbers listed on lines 88 (total n=38) and 96-97 (15 flat roamers, 15 down roamers). Also, this conflicts with the statement on lines 393-394 ("Excluded from the subgroup analysis are 2 mice with in-between slope values and 6 mice with higher than 0.006 slope values").

This has been corrected. See also comments above.

2) Line 165: do you mean Figure 2B (as opposed to 3B)?

This has been corrected.

3) Line 175: I would suggest averaging over cortical and subcortical regions separately, given their distinctive function, development, gene expression, connectivity, and covariance patterns.

We did not understand, what exactly the reviewer would like to see at this point and why. It is correct that these major brain regions differ in many regards, but what would be gained here by the separate analysis?

4) Line 282: missing closing parenthesis.

This has been corrected.

5) There are various places in the manuscript where structural covariance is used as a proxy to traditional measures of connectivity. While structural covariance has been found to be statistically associated structural and functional connectivity, the idea that structural covariance is driven by synchronized plasticity between structurally connected regions has yet to be confirmed. Therefore, I would caution against making this connection (no pun intended) between covariance and connectivity. A mention of this (and other study limitations) in the discussion might be helpful.

See above. We think that pointing to the hypothesis is useful. We have added a clarifying caveat to the discussion and the introduction.

6) Lines 550-551: Could you double check the definition of the p-value? Unless NBS (which I am admittedly not an expert on) defines the p-value differently, the p-value is a probability of observing data given the null, as opposed to a probability of an effect.

This is an embarrassing mistake that has been corrected.

7) Reasons for only including female mice should be explicitly listed – I assume male mice were excluded to avoid aggression/mating?

This information was added.

8) Figure 3: it would be helpful to annotate the rows and columns of these matrices with names of coarse structures.

As we do not want to imply conclusions about particular associations and the graphs got quite cluttered with the additional label we have refrained from doing so. We are comparing the gross patterns. In higher-resolution follow-ups that are powered to detect more differences we will provide the additional information.

Potential ideas to further improve the study include:1) Are anatomical changes driven by neurogenesis? I.e., does the size of the dentate gyrus correlate with DCX?

Our study was not powered to address this point.

2) Apart from access to a larger and richer space, the ENR group differs from STD in the number of mice and therefore potential social interactions. I'd be curious to know whether these results can be separated into social vs spatial exploration effects. For example, based on RFID data, do flat roamers tend to spend time with other mice more, while down roamers are more isolated? If it is possible to extract from the RFID data, I think an analysis that includes measures of social interactions would make this a far stronger study.

In principle yes, but not yet for the current study. The extraction of social information from the antenna data is an ongoing project and not trivial. We will need a higher-powered study and different readouts to optimally use this information.

3) Similar to the above comment, can the effect of activity be teased out from the RFID data? Exercise has been shown to correlate with brain structural differences; I wonder if some of the neuroanatomical variation seen comes about from mice just moving around more.

Raw antenna counts give a much lower correlation, even if corrected for unique antenna contents (that is to dismiss repeated signals from the same antenna). RE adds the component of territorial coverage, which seems to add to the contribution of “mere activity” (at least as measured by unique antenna contacts). We are currently working on extracting more data, e.g. velocity, directionality, etc. to further qualify “activity”.

Reviewer #3 (Recommendations for the authors):– The interpretation of structural covariance is oversimplified especially in introduction. Specifically, REF 19 is being mischaracterized – this paper did suggest some convergence but also significance divergence with white matter connectivity. In the discussion the authors invoke the idea that structural covariance "may reflect synchronized plastic changes occurring across multiple brain regions over time". This idea should be included in the motivation for the investigation as well.

We have clarified this point.

– Given references to prior related work by the authors and others, it is not currently clear what are the exact analyses that have not previously been done which are unique to this paper. The paper would benefit from a clear discussion paragraph stating what this study does that has not been done previously and what is a replication (differentiating replication from restatement of results already reported in the same mice).

The study uses the same general set-up as Zocher et al. 2019 and 2020, but is otherwise unique. No MRI was used in those studies. We have clarified, which of the other results are confirmative.

– The statement on p.7 that "The majority of these ROIs were conserved as significant group differences when considering relative volumes (% of whole brain), suggesting normal scaling" should be clarified. Do the authors mean normal scaling versus nonlinear scaling cf https://www.science.org/doi/10.1126/science.aar2578? Note that dividing by total brain size is not really an effective way to "control" for the effect of total brain size (https://www.sciencedirect.com/science/article/pii/S1053811922006012)

We thank the reviewer for raising this point and we agree with the view that a significant proportion of the variance in the volume of many (but not all) brain structures is accounted for by overall differences in brain volume. Hence, using relative volumes will lower the standard deviations and may be more sensitive to subtle anatomical changes. In the context of the current study, we analyzed both absolute and regional volumes, the latter expressed as a percentage of the total brain volume. We note that the main statistically significant findings (5% FDR) regarding neuroanatomical differences between STD and ENR mice and the exploratory findings (p<0.01 uncorrected) comparing “down” vs. “flat” roamers were essentially unchanged, i.e. a very similar pattern of structures were identified. We therefore decided to only include absolute volumes in the current paper for the sake of simplicity. For ease of comparison, we include Author response image 1 to highlight the similarities in the results when comparing STD and ENR mice, using either atlas based segmentation or voxel-wise tensor based morphometry.

**Author response image 1. sa2fig1:** Neuroanatomical differences in absolute volumes between STD-ENR comparing absolute volumes for both atlas based segmentation (a) and voxel-wise tensor based morphometry (b), to the same analyses using relative volumes where inter-individual variance in mouse total brain volume is accounted for (c, d). In panels (a) and (c), the highlighted regions shown pass statistical significance at a threshold of 5% FDR for multiple comparisons, with the colour bar indicating the effect size as standardized mean difference. In panels (b) and (d) only voxels that pass a threshold of FWE p<0.05 are shown with the colour bar indicating the % volume difference between the groups. In both cases the great majority of effects are conserved with similar effect sizes, particularly for the hippocampus a mouse brain region known to show a high sensitivity to allometric effects.

– The chi-square tests for equality of two correlation matrices would be better if replicated with permutation testing as REF 21 seems to indicate these tests are anti-conservative in small samples.– NBS uses permutation testing which is good. However it is more rigorous to consider the t-statistic over a range of thresholds not just a single threshold. I'd suggest 2.5-3.5 in 0.1 intervals as a sensitivity analysis. Also it's not clear why t=2.4 is mentioned in results but t=3.1 in mentioned in the methods.

Please see comment above and the added panels in Figure 4. T=2.4 is the threshold at which the network is visualized in the figure, while T=3.1 is the threshold of the upper end of the spectrum of tested thresholds as now also depicted in panels C and D. A clarifying sentence has been added.

– This is out of my area of expertise but is it possible that genetic differences including de novo mutations could underlie some inter-individual differences between inbred mice?

de novo mutations in C57BL/6J have been measured to occur at a rate of 5.4 x 10-9 per nucleotide per generation (Uchimura et al. 2015, Genome Res. 25, 1125–1134), which is too low to explain a reproducible pattern as observed in our studies.

– I am having trouble understanding the rationale for the precise cut off values between groups. On p.15 manuscript says "Excluded from the subgroup analysis are 2 mice with in-between slope values and 6 mice with higher than 0.006 slope values." while on p.10 it says "Mice that showed in-between slope values (n = 10) were excluded from the clustering to achieve a sharper distinction. Presumably, I am not understanding something but it should be more clear.

This confusing statement has been clarified. Please refer to our comments on that issue to referee #1.

– In addition, the rationale for excluding "up roamers" with higher slope values is not clear. These mice should be included in the flat group in a sensitivity analysis to assess robustness of findings.

Please refer to our comments on that issue to referee #1. We have now included the graphs depicting the actual slopes in Figure 3B. We in addition now present the continuous calculation in Figure 2.

– Currently it is not clear to what extent findings depend on the specific thresholds chosen to differentiate flat vs down roamers. It would be an improvement to conduct a sensitivity analysis in order to confirm the robustness of the findings to methodological variability to methodological choices which may seem arbitrary to a reader.

See above for the rationale to use the stratification. We now include the correlation along the entire range of RE slopes as well.